# Risk Perception of Health Risks Associated with Radiation Exposure among Residents of Okuma, Fukushima Prefecture

**DOI:** 10.3390/ijerph182413208

**Published:** 2021-12-15

**Authors:** Keiko Oishi, Makiko Orita, Yasuyuki Taira, Yuya Kashiwazaki, Hitomi Matsunaga, Noboru Takamura

**Affiliations:** Department of Global Health, Medicine and Welfare, Atomic Bomb Disease Institute, Nagasaki University Graduate School of Biomedical Sciences, Nagasaki 8528523, Japan; tak011004@gmail.com (K.O.); y-taira@nagasaki-u.ac.jp (Y.T.); y-kashiwazaki@nagasaki-u.ac.jp (Y.K.); hmatsu@nagasaki-u.ac.jp (H.M.); takamura@nagasaki-u.ac.jp (N.T.)

**Keywords:** Fukushima, nuclear power plant accident, Okuma, radiation exposure, risk perception

## Abstract

Ten years have passed since the Great East Japan Earthquake and the subsequent Fukushima Daiichi nuclear power plant accident on 11 March 2011. Okuma is a town hosting the Fukushima Daiichi nuclear power plant. The evacuation order for Okuma was partially lifted in April 2019. To clarify factors associated with risk perceptions of radiation among the residents of Okuma, we conducted a questionnaire survey in January 2021. Our results revealed that resident anxieties regarding the health effects of radiation exposure from living in Okuma were independently associated with positive PCL-Specific score, recognition of the consultation services with radiation experts in the municipal government of Okuma, and planned request for consultation service regarding radiation exposure by radiation experts, along with being female and living with a child. It is important for radiation experts to promote periodic communication of risks with individuals on a small scale to provide accurate information about the health effects of radiation and to provide maternal and child healthcare services and support regarding child-rearing and radiation exposure, to reduce concerns about radiation exposure and facilitate healthy living and wellbeing in Okuma.

## 1. Introduction

Ten years have passed since the Great East Japan Earthquake and the subsequent Fukushima Daiichi nuclear power plant (FDNPP) accident on 11 March 2011. This accident was the worst nuclear disaster since the Chernobyl nuclear power plant accident in 1986 [1]. On 11 March 2011, a 9.0-magnitude earthquake struck the east coast of Japan. This earthquake caused a tsunami with a height over 15 m to hit the FDNPP, leading to core meltdowns in Reactors 1, 2, and 3 and hydrogen explosions in Units 1, 3, and 4 in the following days. Consequently, radionuclides from the damaged plant were released into the environment. An order to evacuate or remain inside was issued to local residents by the Prime Minister of Japan in his role as the Director-General of the Nuclear Emergency Response Headquarters. At 20:50 that day, residents living within 2 km of the plant were ordered to evacuate. This order was extended to a 3-km radius at 21:23 that day, then to a 10-km radius on the morning of 12 March, and finally to a 20-km radius that afternoon [2]. These decisions have resulted in residents being forced into a long-term, widespread evacuation. Okuma Town is the site of Units 1–4 of the FDNPP. Before the accident, the population of this town was 11,505, but the town was totally evacuated in the aftermath of the accident at the FDNPP.

The long-term effects of radiation need to be evaluated. The Fukushima Health Survey conducted by Fukushima prefectural government estimated the external radiation exposure dose received by Fukushima residents based on their behavior during the four months after the accident (11 March to 11 July) [3]. This survey indicated individual external doses for 466,639 residents of Fukushima Prefecture. Of these, 290,398 (62.2%) received <1 mSv; 147,496 (93.8%) received <2 mSv; and 25,770 (99.4%) received <3 mSv [3]. In cases of exposure to more than 100 mSv of radiation, both the incidence of cancer and the death rate increased with exposure doses [4]. Based on such evidence, the International Commission on Radiological Protection (ICRP) has recommended that the public be exposed to no more than 1 mSv of radiation per year under normal conditions [5]. Even during radiation emergencies like the Fukushima accident, the ICRP recommends limiting annual exposure to radiation, as far as possible, to within the range of 20–100 mSv/year. Moreover, after the accident itself was over, the ICRP recommended that the dose level to optimize protection from radiation for individuals living in contaminated areas should be within the lower range of 1–20 mSv/year. The Fukushima Health Survey also included individual external doses for 4814 residents of Okuma; of these, 3374 (70.1%) received <1 mSv; 1284 (96.8%) received <2 mSv; 112 (99.1%) received <3 mSv; and 23 (99.6%) received <5 mSv [6]. The World Health Organization released a report in 2013 and reported that the disaster would not cause any observable increase in cancer rates in the region [7]. In March 2021, the United Nations Scientific Committee on the Effects of Atomic Radiation (UNSCEAR) summarized that no adverse health effects directly related to radiation from the disaster had been documented among residents of Fukushima. Any future radiation-related health effects were unlikely to be discernible [8].

Evacuation Order areas have been rearranged by the Japanese government [9]. Since 2013, residents of Okuma have been allowed to enter the town only during daylight hours. Based on the lower radiation levels achieved through decontamination work in parts of the town, the government decided to partially lift the Evacuation Orders in April 2019. This decision allowed residents to return to the town. As of January 2021, only 285 people (3.5%) of the original population were registered as returnees [10]. Some residents have presumably decided to never return out of a fear of radiation, have built new lives elsewhere, or do not want to return to where the disaster hit. Risk perception concerns the subjective judgments that individuals make regarding the characteristics and severity of risks. Such judgments are often shaped by the personal experiences of individuals, the news media, and cultural worldviews, via two factors: “dread risk” and “unknown risk” [11,12]. Dread risk refers to hazards that individuals perceive as lacking controllability and carrying risks of catastrophic or fatal consequences, whereas unknown risk pertains to hazards that people perceive as new and that may exhibit delayed manifestations of harm. Nuclear power plant accidents are characterized by strong dread and unknown risk and are thus considered to provoke high risk perception [12,13,14]. However, no research has been conducted on risk perceptions of radiation exposure among residents of Okuma. Risk perception of the health risks associated with radiation exposure needs to be evaluated in residents to implement comprehensive risk communication strategies. The purpose of this study was, therefore, to clarify the factors related to radiation exposure among the residents of Okuma.

## 2. Materials and Methods

### 2.1. Participants

This study was conducted in Okuma town, Fukushima Prefecture in January 2021. The subjects of this study were former residents of Okuma who had resident cards as of 11 March 2011 and still had them in November 2020. We initially distributed questionnaires to residents who were >20 years old. We obtained responses from 1225 residents. After excluding 91 residents with insufficient responses, responses from 1134 residents were analyzed. All study protocols were approved by the ethics committee of Nagasaki University Graduate School of Biomedical Sciences (approval no. 20060103-2).

### 2.2. Questionnaire

The questionnaire for this study was developed based on a questionnaire used in previous studies conducted in Fukushima Prefecture [15,16] and on the mental health and lifestyle survey within the framework of the Fukushima Health Management Survey, which was organized by the Fukushima prefectural government [3,14,17]. We asked about demographic variables including sex, age at the time of the study, birthplace, family members, and living with children aged <18 years. We also asked about intentions to return home, daily physical activity, recognition of the consultation services with radiation experts in the municipal government of Okuma, and planned requests for consultation services regarding radiation exposure by radiation experts. We included questions to evaluate risk perceptions of the health risks associated with radiation exposure in the survey, such as the health effects of radiation in children and on the next generation, whether the resident has anxiety about consuming locally-produced foods in Okuma, and whether the resident has anxiety about the health risks of radiation exposure while living in Okuma. These questions were evaluated using a four-point scale (1 = yes, 2 = probably yes, 3 = probably no, and 4 = no).

Psychological distress was assessed using a post-traumatic stress disorder (PTSD) checklist (PCL). The PCL is a self-administered questionnaire widely used to assess the severity of traumatic reactions and to screen for those with a diagnosis of PTSD. The psychometric and screening properties of the PCL have been reported [18,19]. Among several versions of the PCL, the PCL-Specific version (PCL-S) is applied to individuals who have experienced specific traumatic events. We used an abbreviated version of the PCL-S, consisting of the following 4 items: “intrusive recollections”; “reaction to re-minders”; “avoidance of reminders”; and “concentration difficulties”. The validity of the PCL-S has been confirmed by the Fukushima Health Management Survey [20]. Participants indicated whether they were bothered by symptoms due to the traumatic event in the past month on a 5-point Likert scale. (1 = not at all to 5 = extremely), with the sum of scores ranging from 4 to 20. We used a standard cutoff of ≥12 to indicate mood/anxiety disorders, as applied in previous Japanese studies [20].

### 2.3. Statistical Methods

We identified factors associated with risk perception for the health effects of radiation using the chi-square test. We also conducted logistic regression analysis and calculated odds ratios (ORs) to identify risk perceptions regarding the health effects of radiation. Data were statistically analyzed using IBM SPSS Statistics version 25 software. Values of *p* < 0.05 were considered statistically significant.

## 3. Results

Of 1134 residents, 642 (57%) answered that they felt anxiety about the health effects of radiation exposure due to living in Okuma (Anxiety (+)). The remaining 492 residents (43%) reported no anxiety about the health effects of radiation exposure (Anxiety (−)). Women were significantly more frequent among Anxiety (+) than Anxiety (−) residents, as were residents living with children. Age, number of family members, birthplace, daily physical activity, and thinking that life is worth living did not differ significantly between Anxiety (+) and Anxiety (−) groups (Table 1).

The following percentages were significantly higher among Anxiety (+) residents than among Anxiety (−) residents: those who had concerns about eating food produced in Okuma (83% vs. 21%, *p* < 0.01) and about drinking tap water in Okuma (91% vs. 30%, *p* < 0.01); those who had concerns about the genetic effects on the next generation (88% vs. 19%, *p* < 0.01); those who do not know that the municipal government of Okuma provides a consultation service with radiation experts (58% vs. 49%, *p* < 0.01); and those who plan to request consultations with radiation experts (36% vs. 19%, *p* < 0.01) (Table 2).

Logistic regression analysis revealed that being female (OR 0.50, 95% confidence interval [CI] 0.39–0.64), living with a child (OR 1.53, 95%CI 1.10–2.09), ≥12 of PCL-S (OR 0.62, 95%CI 0.42–0.92), knowing about the consultation services with radiation experts in the municipal government of Okuma (OR 0.67, 95%CI 0.52–0.85) and planned requests for consultation with radiation experts (OR 2.33, 95%CI 1.76–3.10) were independently associated with the anxieties of residents about the health effects of radiation exposure while living in Okuma (Table 3).

## 4. Discussion

This study examined the characteristics and factors related to risk perception for the health effects of radiation among residents of Okuma, the host of Units 1–4 of the FDNPP. Our results showed that 57% of residents had some anxiety regarding the health risks associated with radiation exposure while living in Okuma (Anxiety (+) group), whereas 43% of residents had no such anxiety (Anxiety (−) group). The 2011 Fukushima Health Management Survey revealed that 48% of Fukushima residents believed that they would experience health effects from radiation [17,21], whereas this percentage had gradually decreased to 29% in the 2019 Fukushima Health Management survey [21]. The risk perception of Fukushima residents regarding radiation exposure has thus been gradually improving. On the other hand, in Okuma, this issue has not been considered. This result might be due to the marked difference in the extent to which each area has progressed in terms of the phase of recovery. In the case of the village of Kawauchi, which is located less than 30 km from the FDNPP and was partially included in the Evacuation Order Area established within a 20-km radius from the FDNPP, the mayor of the village declared that residents could return to their homes in January 2012, after the Japanese Prime Minister had declared that the FDNPP reactors had achieved a state of “cold shutdown” in December 2011 [22]. A decade has passed since the FDNPP accident, and about 2460 of the original 2700 residents of Kawauchi have returned. Everyday life has largely been restored, allowing life in the village to basically return to what it used to be. On the other hand, Tomioka, a town neighboring Kawauchi and Okuma, had 15,937 residents before the accident. Of those, only around 1800 residents have returned since the evacuation order was lifted in April 2017 [23], and Tomioka is still in the process of recovering. Meanwhile, about 300 residents have returned to Okuma. This means that Okuma has just started on the path to recovery. The needs of each area have differed, and with respect to such areas, providing support according to the needs of the different phases of recovery is important in each affected municipality.

In this study, the chi-square test showed that the following percentages were significantly higher among Anxiety (+) individuals than among Anxiety (−) individuals: residents who had concerns about eating food produced in Okuma (83% vs. 21%, *p* < 0.01); residents who had concerns about drinking tap water in Okuma (91% vs. 30%, *p* < 0.01); and residents who had concerns about genetic effects on the next generation (88% vs. 19%, *p* < 0.01). Our previous study conducted in Kawauchi village in 2014 also indicated a markedly bipolar nature of the risk perception of the health effects of radiation among residents after the FDNPP accident and suggested such serious misunderstandings of radiation and its health effects might lead to distress and anxieties from a loss of livelihoods [15]. Such tendencies in the perceptions of radiation risk among residents may be observed regardless of the timing of evacuation orders being lifted and might lead to major impacts on mental health stress and social well-being. Our study also indicated that women and those living with a child expressed greater concerns about the health effects of radiation exposure. We may surmise that women, particularly those of childbearing age, may have particular concerns about the possible effects of radiation on fertility and progeny [24]. Ensuring safety for women of childbearing age by enhancing maternal and child healthcare services that provide consultation and support regarding child-rearing and radiation exposure is essential.

In addition, this study showed that the frequencies of positive PCL-S were significantly higher among Anxiety (+) individuals (15%) than among Anxiety (−) individuals (9%), despite the frequency of positive PCL in 8.3% from the 2019 Fukushima Health Management Survey [21]. Resident anxiety regarding the health effects of radiation exposure while living in Okuma appears associated with a high prevalence of post-traumatic stress. Mental health problems such as depression and PTSD represent important public health issues in the long term after the nuclear accident. Previous studies conducted in Chernobyl and Fukushima have also indicated that risk perception for radiation exposure was associated with poor mental health [14,17,25,26]. The Fukushima Health Management Survey revealed that psychological distress was associated with higher risk perception in the early years after the accident [14,17]. Suzuki et al. suggested that a strong initial risk perception can have a strong impact, particularly on how residents obtain information related to the risk [17]. If the obtained information is consistent with their original beliefs, residents tend to believe in the accuracy of that information, whereas if not, residents might dismiss the information as untrustworthy. To reduce the anxiety and mental health stress received in the early phase of the accident, continuous support for life-planning and radiation protection needs to be provided to residents in cooperation with the relevant local governments and experts.

Since the evacuation order was lifted, individual consultations with radiation experts have been provided at the municipal government of Okuma. On the other hand, our results revealed that the frequency of residents who were not yet aware of consultation services with radiation experts were significantly higher in Anxiety (+) residents than in Anxiety (−) residents. In addition, the frequency of residents who planned to request consultations with radiation experts was significantly higher among Anxiety (+) residents than among Anxiety (−) residents. The Fukushima Health Management Survey in 2016 found that 3650 of 32,699 respondents (11.2%) answered they had no one or never went to a specific institution for consultations when experiencing mental or physical problems following the Great East Japan Earthquake [27]. Mental instability due to evacuation might have been a common reaction [27,28]. Due to the long-term nature of the evacuation, the social networks and connections of residents with the local authority may have been disconnected. The results suggested that poor connectivity, such as reduced social networks result in fewer opportunities to obtain information that residents want to know despite help from many local care providers. Our results also showed that residents who had anxiety about the health effects of radiation exposure desired consultations with radiation experts. Measures that allow residents to consult without hesitation need to be promoted. After the FDNPP accident, various public communications from the prefectural level to the individual level have been implemented [29], such as a general health consultation project through Fukushima Health Management Surveys [29]. These activities generally started with radiation risks, mainly through group-based discussions, but with the passage of time, gradually shifted to face-to-face communications to address comprehensive health risks to individuals and well-being [29]. Promoting overall public health for residents of Okuma by encouraging individual engagement and establishing social relationships through risk communication is essential.

This questionnaire study was conducted for the first time among the residents of Okuma, a town hosting part of the FDNPP. Nevertheless, this study showed several limitations. First, the establishment of a cause-and-effect relationship was not possible because this study used a cross-sectional study design. We recommend that more longitudinal studies be conducted to establish causal relationships. In addition, Okuma includes a difficult-to-return-to area for which the Evacuation Order has yet to be lifted, which may have caused bias in the residential areas and the intentions of ex-residents to return to their original homes.

## 5. Conclusions

Our results revealed that resident anxieties about the health effects of radiation exposure while living in Okuma were independently associated with positive PCL-S, recognition of the availability of consultation services by radiation experts, and requests for consultations with radiation experts, as well as being female and living with a child. It is important for radiation experts to develop risk communication with people on a small scale periodically for providing accurate information about the health effects of radiation and to provide the maternal and child health care services that provide consultation and support for child-rearing and radiation exposure in order to reduce their concerns about radiation exposure and to help their healthy living and well-being.

## Figures and Tables

**Table 1 ijerph-18-13208-t001:** Demographic factors and PCL-S scores of residents by risk perception for health effects of radiation exposure while living in Okuma.

Variables	Reference	Anxiety (+) (*n* = 642)	Anxiety (−) (*n* = 492)	*p*
Sex	Male	265 (41%)	290 (59%)	<0.01
Age	<60 years	226 (35%)	171 (35%)	0.90
Family members (including self)	Alone	107 (17%)	97 (20%)	0.07
	Two	282 (44%)	233 (47%)	
	Three or more	253 (39%)	162 (33%)	
Living with a child < 18 years old	Yes	137 (21%)	78 (16%)	0.02 *
Born in Okuma	Yes	347 (54%)	270 (55%)	0.78
Physical activity for >1 h/day	Yes	356 (56%)	289 (59%)	0.27
Intention to return to Okuma	Already returned	9 (1%)	29 (6%)	<0.01 *
	Want to return	43 (7%)	68 (14%)	
	Hard to judge	177 (28%)	99 (20%)	
	Will not return	413 (64%)	296 (60%)	
PCL-S score	≥12	98 (15%)	44 (9%)	<0.01 *
Life is worth living	Yes	409 (64%)	335 (68%)	0.13

* Significant difference according to the chi-squared test.

**Table 2 ijerph-18-13208-t002:** Risk perception about health effects of radiation exposure.

Variables	Reference	Anxiety (+) (*n* = 642)	Anxiety (−) (*n* = 492)	*p*
Concerns about eating food produced in Okuma Town	Yes	532 (83%)	104 (21%)	<0.01 *
Concerns about drinking tap water in Okuma Town	Yes	583 (91%)	146 (30%)	<0.01 *
Concerns about genetic effects in the next generation	Yes	567 (88%)	93 (19%)	<0.01 *
Awareness of consultation services with radiation experts	Yes	271 (42%)	253 (51%)	<0.01 *
Planned requests for consultation with radiation experts	Yes	229 (36%)	94 (19%)	<0.01 *

* Significant difference according to the chi-squared test.

**Table 3 ijerph-18-13208-t003:** Logistic regression analysis for resident anxiety about the health effects of radiation exposure.

Variables	Reference	OR	95%CI
Sex	male/female	0.50 **	0.39–0.64
Living with a child aged < 18 years	yes/no	1.52 *	1.10–2.09
PCL-S score	<12/≥12	0.62 *	0.42–0.92
Recognition of consultation services with radiation experts	aware/unaware	0.67 **	0.52–0.85
Consultation with radiation experts	yes/no	2.33 **	1.76–3.10

OR, odds ratio; CI, confidence interval. * *p* < 0.05; ** *p* <0.01, logistic regression analyses.

## Data Availability

All data are available from the corresponding author on reasonable request.

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
