# Peer review of "Risk Perception of Health Risks Associated with Radiation Exposure among Residents of Okuma, Fukushima Prefecture"

_ijerph, 2021, doi:10.3390/ijerph182413208_

Round 1

Reviewer 1 Report

The authors present the results of a survey about a significant sample of Okuma's population of 1,225 people. The main results of this survey are very clearly presented just like the method which is traditionally used for this type of study (Chi-square, logistic regression). I would also like to thank the authors for the clarity of their remarks because the article is easy to read. The reminder of the facts in the introduction and in particular of the chronology of the evacuation perimeters orders were particularly appreciated.

-Regarding the form, a ">" is missing in line 75.

-We would have liked to get the details of the questions asked (questionnaire deposited as a supplementary material) knowing that the way in which questions are formulated could influence the results.

-The bibliography is a little too Japanese centered, it would be good adding the following references:

Venables et al., 2009

Venables D., Pidgeon N., Simmons P., Henwood K., Parkhill K.

Living with nuclear power: A Q-method study of local community perceptions

Risk Anal., 29 (8) (2009), pp. 1089-1104

Slovic et al., 2004

Slovic W., et al.

Risk as analysis and risk as feelings: Some thoughts about affect, reason, risk, and rationality

Risk Anal., 24 (2004), pp. 311-322

Stoutenborough et al., 2013

Stoutenborough J.W., Sturgess S.G., Vedlitz A.

Knowledge, risk, and policy support: Public perceptions of nuclear power

Energy Policy, 62 (2013), pp. 176-184, 10.1016/j.enpol.2013.06.098

Reviewer 2 Report

Thank you for the opportunity to read and review your paper. The authors assess risk perception of the health effects associated with radiation exposure in Okuma, Japan ten years after the Fukushima disaster. Better understanding what factors shape risk perception is an important topic and the paper is clearly written, but I have still some concerns that should be addressed before publication.

  • One of main issues lies with the fact that the analysis and conclusions remain vague and superficial. The authors do not explore specifically in which ways their findings can help inform risk perception and mental health management. For instance, they mention that being female and having a child in one’s household are associated with greater anxiety towards radiation exposure, yet they don’t address how public health efforts can reflect this. Despite asserting that promoting public health and well-being is important – which they are – the authors don’t clearly define what potential outcomes could come from their findings. Overall, I suggest that the discussion and conclusions should provide a more robust and less vague analysis of the results considering ways to implement public health measures in communities affected by health-related anxieties.
  • Similarly, the discussion section includes rather long overviews of past studies. It is fine to use past studies to compare and contrast the authors’ current findings, but it seems to me that more time is dedicated to discuss these past studies than to explore the outcomes of the findings presented in this paper.
  • On page 5 (lines 171-178), the authors talk about the bipolarization of risk perception, I wonder if this is because of the scale used in the study? I suggest that the authors make sure that they address this aspect of data analysis.
  • Some statements in the paper would benefit from additional details. For instance, on page 1 (lines 43-44), the authors mention the estimated external radiation dose received by Fukushima residents but do not indicate what the value or the level of risk it represents. Similarly, on page 2 (lines 46-47) they point out the different external doses received by Okuma residents without including the reference dose for comparison. One pages 2-3 (lines 94-101), the authors discuss the PCL without indicating how it is calculated or discussing the cutoff. Providing your readers these reference frames would strengthen the paper.
  • The discussion of the different Evacuation Order areas on page 2 (lines 54-69) is a little long and, as is, does not seem to be utilized later in the paper. I recommend shortening this section to maybe better address what risk perception entails – since it is one of the important aspect of the study – and how it is shaped and reshaped. 
  • While the paper is clear and overall well-written, double-checking for repetitive, vague, or cumbersome statements like "female sex" – would it be better to talk about “health risks associated with radiation exposure even though it adds an extra word? – would strengthen the argument.

Round 2

Reviewer 2 Report

The authors have addressed the concerns I had with the paper. Their revisions  have strengthened the argument. Thank you!